# Beyond Purple Tomatoes: Combined Strategies Targeting Anthocyanins to Generate Crimson, Magenta, and Indigo Fruit

Eugenio Butelli [1,*] , Katharina Bulling [1,2], Lionel Hill [1] and Cathie Martin [1]

1 John Innes Centre, Norwich NR4 7UH, UK; Katharina.bulling@gmail.com (K.B.); lionel.hill@jic.ac.uk (L.H.); cathie.martin@jic.ac.uk (C.M.)
2 Danone Place Amsterdam, Taurusavenue 167, 2132 LS Hoofddorp, The Netherlands
* Correspondence: eugenio.butelli@jic.ac.uk

**Abstract:** The range of colours of many flowers and fruits is largely due to variations in the types of anthocyanins produced. The degree of hydroxylation on the B-ring affects the hue of these pigments, causing a shift from the orange end of the visible spectrum to the blue end. Besides colour, this modification can also affect other properties of anthocyanins, including the ability to protect the plant against different stresses or, when included in the human diet, to provide benefits for disease prevention. The level of hydroxylation of the B-ring is determined by the activity of two key hydroxylases, F3′H and F3′5′H, and by the substrate preference of DFR, an enzyme acting downstream in the biosynthetic pathway. We show that, in tomato, a strategy based on fruit-specific engineering of three regulatory genes (*AmDel*, *AmRos1*, *AtMYB12*) and a single biosynthetic gene (*AmDFR*), together with the availability of a specific mutation (*f3′5′h*), results in the generation of three different varieties producing high levels of anthocyanins with different levels of hydroxylation. These tomatoes show distinctive colours and mimic the classes of anthocyanins found in natural berries, thus providing unique near-isogenic material for different studies.

**Keywords:** anthocyanins; tomato; fruit colour; transcriptional regulation; LC-MS/MS





## 1. Introduction

Anthocyanins are strikingly vibrant natural plant pigments responsible for a wide range of colour variation—from orange to blue—in many flowers, fruit, and vegetables. In addition to their well-documented physiological roles in plant life, which include attraction of pollinators and seed dispersers and protection against biotic and abiotic stresses [1], dietary anthocyanins have been linked to a wide range of health benefits and protection against the most prevalent chronic human diseases (cardiovascular disease, cancer, diabetes, obesity, and neurological disorders) [2–4]. Although much focus has been given to their chemical reactivity and in vitro free-radical scavenging capacity [5,6], growing evidence indicates that other mechanisms are responsible for the observed in vivo health benefits, and more recent studies suggest critical roles in the modulation of different signalling pathways [3] and the interaction with the gut microbiota [7,8].

Despite several decades of multidisciplinary research, the complexity of the molecular mechanisms underpinning the health benefits of anthocyanins remain largely undiscovered. The chemical diversity and complexity of anthocyanins is a major obstacle. Nearly one thousand different naturally occurring compounds have been reported, whose structures differ in the combinations of sugar units, methoxyl groups, aliphatic and aromatic acyl groups decorating the basic structure of anthocyanins: a C6-C3-C6 skeleton structure consisting of two aromatic rings (A and B) and a central heterocyclic oxygen ring (C) with a positive charge (Figure 1), also known as a flavylium cation [9].

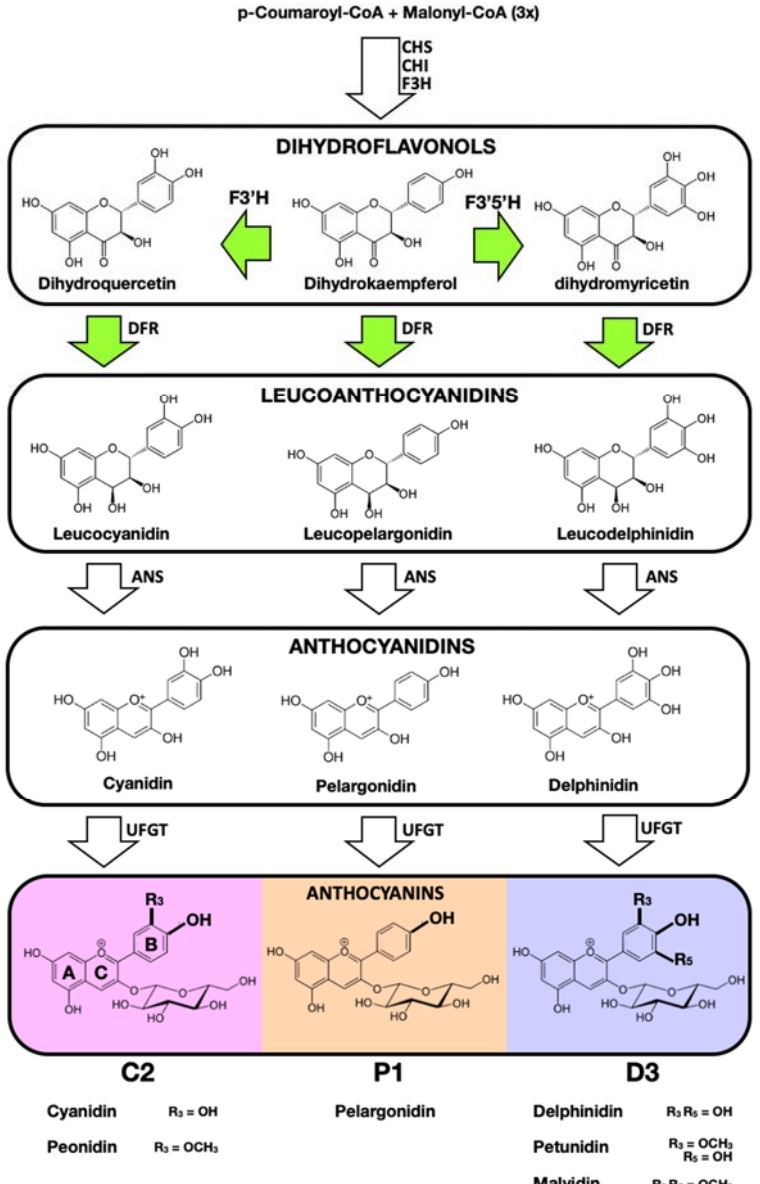

**Figure 1.** Schematic representation of the late stages of the anthocyanin biosynthetic pathway leading to the production of three classes of anthocyanins (here abbreviated as P1, C2 and D3) with different colour properties associated with the pattern of hydroxylation of the B-ring. Key genes required for the differential accumulation of P1, C2 or D3 are indicated by green arrows.

The hydroxylation pattern of the B-ring has a major influence on colour and is associated with a shift from orange/red to violet/blue with an increasing number of hydroxyl groups [9,10]. The final colour of plant tissues is determined by a combination of different factors (including vacuolar pH, the presence of co-pigments and metal ions, and the shape of the cell) but the B-ring hydroxylation pattern has a major impact. This pattern can influence the physiological and ecological functions in plants, affecting pollinator preference for specific flower colours [11] or protection against specific pathogens and UV light [12]. The level of B-ring hydroxylation also impacts on the antioxidant activity of dietary anthocyanins [13], which has been associated with their bioactive potential.

The division of anthocyanins into three main classes based of the hydroxylation pattern of the B-ring attests to the importance of this structural modification. The hydroxyl group at the 4′ position of the B-ring is usually present because its addition (from coumaroyl CoA) occurs early in the pathway, leading to the formation of the chalcone backbone,

common to almost all classes of flavonoids. The mono-hydroxylated anthocyanin structure is named pelargonidin (here abbreviated as P1). Positions 3′ and 5′ can be hydroxylated by the action of specific cytochrome P450 enzymes, flavonoid 3′-hydroxylase (F3′H) and flavonoid 3′,5′-hydroxylase (F3′5′H), leading to the formation of di-hydroxylated cyanidin (C2) or tri-hydroxylated delphinidin (D3). Additional methylation at these two positions is also common, giving origin to peonidin (C2-derived), petunidin and malvidin (both D3-derived; Figure 1).

The potential health benefits of the three different classes of anthocyanins have been very difficult to assess. Structure–activity relationship studies using individual compounds are not feasible in vivo, and limited comparative studies are available where different fruit or plant extracts have been tested in human intervention studies or in animal models, because variation in the anthocyanin class accumulated is not available within a single type of fruit or vegetable. Consequently, it is possible that specific classes of anthocyanins provide differential protection against specific chronic diseases, but this has not been tested rigorously in nutritional intervention studies in humans or animal models.

A major obstacle in transferring the study of bioactive phytochemicals to personalised dietary recommendations is the presence of a complex food matrix where a certain compound interacts with a myriad of other, potentially bioactive food components. For anthocyanins, the comparison of different types of berries results in many confounding effects due to the presence of other phytochemicals. The name "berry" is colloquially used to indicate fruits that do not share the same botanical definition in terms of morphology and development and is applied to species that do not have a common ontology, spanning different clades within the Eudicots. For example, blueberry is a true berry belonging to the Asterids, while raspberry is an aggregate fruit in the Rosid clade. Therefore, although many berries accumulate high levels of anthocyanins, different berries display very different metabolic profiles underpinning their bioactivities. Ideally, comparative studies should take into consideration varieties within the same species producing different types of anthocyanins but sharing the same genetic background and general metabolic pattern. Unfortunately, even in species where thousands of years of cultivation, breeding and selection have generated a remarkable phenotypic diversity or when such a diversity naturally exists within closely related species, the patterns of anthocyanin hydroxylation and decoration tend to be similar. Apples are considered the most diverse fruit in terms of available varieties, yet the main red pigment is always cyanidin-3-galactoside (including red-fleshed varieties and relatives in the same genus), with other C2 derivatives as minor constituents [14,15]. In grape, anthocyanins have been thoroughly investigated because of their influence on the quality of red wine. Despite the identification of very complex and diverse patterns, malvidin-3-glucoside is the most abundant compound and P1 derivatives can only be detected in trace amounts in some varieties [16]. On the other end, potato offers an attractive opportunity to study different anthocyanins because of renewed interest in pigmented landraces. Purple- and red-fleshed varieties, mainly accumulating D3 and P1, respectively, have been characterised [17]. However, the complexity of anthocyanin composition, which frequently includes considerable amounts of C2, the variability in concentration in both skin and flesh, and the degradation during cooking and processing represent obstacles. Even in ornamental plants, where flower colour is the most important trait, no more than two classes of anthocyanins are ever present in the same species, despite continuous attempts to introduce novel hues by conventional breeding.

We have shown that, in tomato, tissue-specific expression of two regulatory genes, *AmDel* and *AmRos1*, can induce the accumulation of high levels of D3 anthocyanins in tomato fruit [18]. We now show that combining genetic engineering with the availability of natural mutants can expand the chemical diversity of anthocyanins to generate unique tomato lines producing P1, C2, or D3-based anthocyanins.

## 2. Materials and Methods

### 2.1. Plant Material

Production of 'Purple' (*Del/Ros1*), 'Yellow' (*AtMYB12*) and 'Indigo' (*Del/Ros1; At-MYB12*) tomato plants, expressing different regulatory genes under the control of the fruit-specific E8 promoter, were generated by genetic modification, as described previously [18–20]. Seeds of the VF36 *anthocyaninless* (*a*) mutant, corresponding to *f3′5′h*, were obtained from the seed bank of the Tomato Genetics Resource Center (TGRC) at the University of California at Davis, USA (https://tgrc.ucdavis.edu, accessed on 14 September 2021). Wild type and transgenic plants were grown at 23–25 °C under a 16 h light and 8 h dark cycle. These conditions were maintained in the growth room, where the plants were grown on Murashige and Skoog (MS) agar medium, and in the glasshouse.

### 2.2. Plasmid Constrzuction and Plant Transformation

The coding sequence of AmDFR was cloned in SLJ.E8.ROS, a binary vector based on SLJ7292 containing the tomato fruit-specific E8 promoter, the coding sequence of *AmRos1* and a CaMV terminator as previously described [18]. The coding sequence of *AmRos1* was excised from SLJ.E8.ROS using the restriction enzymes *BamHI* and *SmaI* and replaced with the cDNA from *AmDFR* from snapdragon (GenBank accession: X15536.1) [21] previously amplified with primers designed to introduce *BglII* and *SmaI* restriction sites (Supplementary Table S1). The resulting vector (SLJ.E8.DFR) was transferred to *Agrobacterium tumefaciens* strain LBA4404 by triparental mating and used to transform the *f3′5′h anthocyaninless* (*a*) tomato mutant using conventional procedures [22].

### 2.3. Tomato Crosses and Selection of Plants

'Purple' (*Del/Ros1*) tomatoes were initially obtained in the dwarf tomato cultivar Micro Tom. The transgene was transferred to the genetic background Money Maker by crossing followed by selection for big fruit and tall plants over nine generations and a further backcross, after which the plants were considered indistinguishable from Money Maker. 'Yellow' (*AtMYB12*) tomatoes were obtained in both Micro Tom and Money Maker genetic backgrounds by independent transformations. 'Indigo' (*Del/Ros1; AtMYB12*) tomatoes were developed in both backgrounds by crossing the appropriate 'Purple' and 'Yellow' varieties. 'Pink' tomatoes were generated by crossing Micro Tom 'Purple' plants with the *anthocyaninless (a)* mutant in VF36 background. The F2 generation with the desired genotype (*Del/Ros1; f3′5′h*) displayed small fruit and short plants that were considered Micro Tom-like. 'Crimson' tomatoes were initially obtained by transformation of the *anthocyaninless (a)* mutant in VF36 background with *AmDFR*, followed by crossing with the 'Purple' line in the Micro Tom background. The F2 generation with the desired genotype (*Del/Ros1; AmDFR; f3′5′h*) displayed small fruit and short plants (Micro Tom-like). 'Magenta' tomatoes were generated by crossing 'Crimson' with 'Indigo' plants in Money Maker background followed by three generations of self-pollinations and selection for the desired genotype (*Del/Ros1; AmDFR; AtMYB12; f3′5′h*) showing either small fruit and short plants (Micro Tom-like) or big fruit and tall plants (Money Maker-like). During this process, a 'Crimson' line with big fruit and tall plants (Money Maker-like) was also selected.

Tomato crosses were performed by emasculating the flowers of the female parents before the flower opened up and the stamens reached full maturity. Two days after emasculation, the stamens were removed from the male plants and carefully positioned around the stigma of the female parents. Seeds from the resulting fruit were grown on MS agar medium containing kanamycin (100 mg/L) to select for progeny containing at least one transgene and to facilitate the visual identification of the homozygous *f3′5′h* mutation, which is associated with the absence of anthocyanins in the hypocotyl. The desired combinations of transgenes were confirmed by DNA extraction and PCR analysis using gene-specific primers (Table S1).

### 2.4. Extraction, Quantification and Metabolic Analysis of Anthocyanins

Tomato fruits were harvested at the ripe stage, frozen in liquid nitrogen and ground to a fine powder. A crude extract was prepared by extraction of 200 mg of dried tomato material with $2 \times 5$ mL 80% methanol (*v/v*) containing 1% HCl. After centrifugation for 15 min at $4000\times g$, anthocyanins in the supernatants were measured spectrophotometrically and expressed as mg of cyanidin-3-glucoside equivalent per g dry weight, based on an extinction coefficient of 26,900 and a molecular weight of 449.2. Strawberries (var. Inspire) raspberries (var. BerryWorld Gem) and blueberries (var. Eureka) where purchased from a local supermarket and processed in the same way. For each extract, the absorbance spectrum was determined and peak wavelengths of 513 nm ('Crimson' tomato and strawberry), 527 nm ('Pink', 'Magenta' tomatoes and raspberries) or 540 nm ('Purple', Indigo' and blueberry) were used to measure absorbance.

The identification of individual anthocyanin compound was performed on whole fruit or in manually dissected peel and flesh tissues processed as described above or in juice obtained after blending whole fruit in a coffee grinder followed by centrifugation for 15 min at $4000\times g$ and filtration. The juice samples were run on a Shimadzu Nexera LC system attached to an IT ToF mass spectrometer. Separation was performed on a $100 \times 2.1$ mm 2.6 μ Kinetex XB-C18 column (Phenomenex: Torrance, CA, USA) using the following gradient of acetonitrile (solvent B) versus 0.1% formic acid in water (solvent A), run at 0.5 mL/min and 40 °C: 0 min, 2% B; 0.5 min, 2% B; 3 min, 10% B; 13 min, 30% B; 18 min, 90% B; 18.8 min, 90% B; 19 min, 2% B; 23.1 min, 2% B. Detection was conducted using UV/visible absorbance collecting spectra from 200 to 600 nm, from which extracted ion chromatograms could be taken at appropriate wavelengths for each analyte. The instrument also collected positive electrospray MS, with spectra from $m/z$ 200 to 2000 and MS2 spectra of the most abundant precursors, collected at an isolation width of $m/z$ 3.0, and fragmented at 50% collision energy and 50% gas. The spray chamber conditions included a 250 °C curved desabsorption line temperature, 1.5 L/min nebulising gas, and 300 °C heat block. The instrument was calibrated with sodium trifluoroacetate before use according to the manufacturer's instructions (Shimadzu).

## 3. Results

*Metabolic Engineering and Analysis of Tomatoes Producing Different Classes of Anthocyanins*

Unlike other members of the genus *Solanum*, tomato does not accumulate anthocyanins in its fruit, where the red colour is due to the presence of lycopene and other carotenoids. In the last two decades, however, two different strategies have been used to develop purple, anthocyanin-producing tomatoes [23]: genetic modification [18] and interspecific hybridization with wild relatives [24]. Both approaches rely on the introduction of new regulatory genes encoding transcription factors and result in fruit producing exclusively D3 [25,26] and displaying metabolic profiles comparable to the anthocyanins produced naturally in vegetative tissues [27].

The enzyme required for the production of D3 anthocyanins in tomato, a cytochrome P450-dependent monooxygenase which introduces hydroxyl groups in positions 3′ and 5′ of the B-ring [28], is encoded by the *flavonoid 3′,5′ hydroxylase* (*F3′5′H*) gene (Figure 1). This gene is under the control of the exogenous transcription factors in both types of purple tomato.

To engineer the accumulation of novel anthocyanins in tomato fruit, we suppressed the activity of F3′5′H using a mutant, named *anthocyaninless* (*a*), in which the complete inability to produces anthocyanins had been found, associated with a premature stop codon in the coding sequence of the *F3′5′H* gene [29]. We crossed this mutant with the 'Purple' tomato line obtained by fruit-specific expression of two regulatory genes from *Antirrhinum majus* (snapdragon) encoding the transcription factors Del (bHLH-type) and Ros1 (MYB-type) [18]. In the F2 generation, we selected seedlings with the transgenes but lacking anthocyanins in the vegetative tissue (*Del/Ros*; *f3′5′h*), and we confirmed the presence of a homozygous stop mutation in *F3′5′H* (Figure S1) resulting in a truncated

protein of 170 amino acids, considerably shorter than the wild type protein (512 amino acids) and lacking most of the substrate recognition site. Fruit of these F2 plants, named 'Pink', displayed slightly darker skin, a pigmented vascular tissue and flesh with a subtle shade of red-pink (Figure 2). Further analysis indicated that the content of anthocyanins in 'Pink' (*Del/Ros1*; *f3'f'h*) fruit was approximately 7% of that measured in 'Purple' tomatoes (*Del/Ros1*), and consisted entirely of C2 anthocyanins, with no trace of D3 found in *Del/Ros1* fruit. Interestingly, no P1-type anthocyanins were detectable (Figure S2).

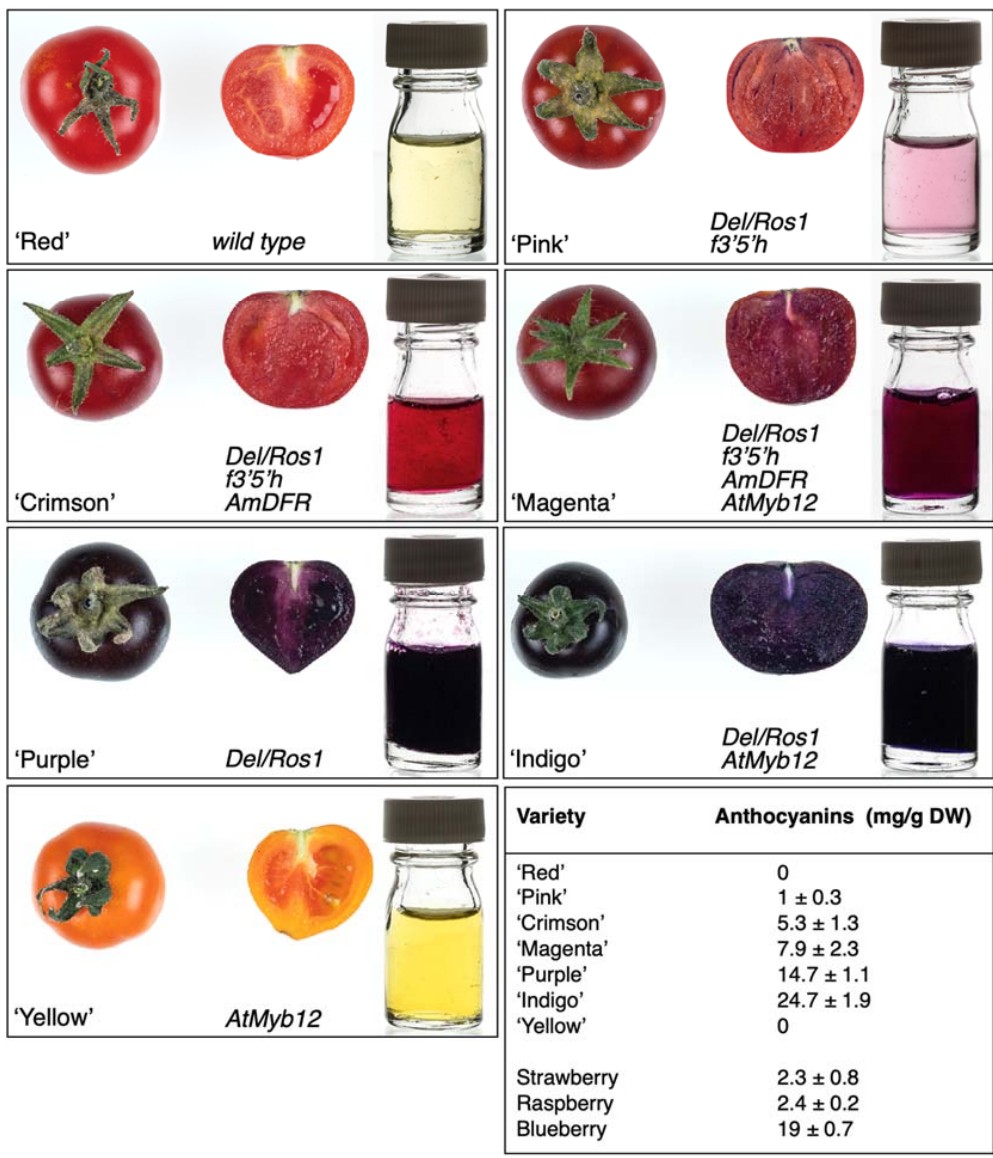

**Figure 2.** Summary of the phenotypes and genotypes of the varieties used in this study in Micro Tom or Micro Tom-like background. Whole and cross-sections of tomato fruit, undiluted tomato juice after centrifugation, genotype and colloquial names are illustrated for each variety. The bottom panel on the right reports the concentration of anthocyanins, expressed as mg per gram of dry weight, in whole fruit; the concentration of anthocyanins in commercial fruit berries is reported for comparison.

Several studies have shown that the synthesis of specific types of anthocyanins is affected by the substrate specificity of dihydroflavonol 4-reductase (DFR) which, in many plants, including Solanaceous species such as petunia and tomato [27,30], displays a strong preference for dihydromyricetin, the precursor of D3 anthocyanins (Figure 1). To overcome the substrate specificity of the endogenous gene, we introduced the *DFR* gene

from snapdragon, a species that can accumulate both P1 and C2 anthocyanins, into the *f3'5'h anthocyaninless* (*a*) mutant. The transgene (*AmDFR*) did not restore anthocyanin pigmentation in vegetative tissues since it was driven by the E8 fruit-specific promoter, and all the T0 plants were indistinguishable from the parental *a* line. Ten plants were crossed with the 'Purple' *Del/Ros1* line and, in the F2 generation, twenty-four plants with the desired genotype (*Del/Ros1; AmDFR; f3'5'h*) were selected. Fruit of most of them displayed darker skin and distinctive novel deep red colour in the flesh and were named 'Crimson' (Figure 2). Analysis of whole tomato fruit indicated that both P1 and C2 anthocyanins were produced. When the same analysis was carried out separately on isolated fruit tissues, P1 was identified exclusively in the fruit flesh, while the peel contained only C2-type anthocyanins (Figure S3).

The gene that determines the partitioning of anthocyanins between P1 and C2 is *flavonoid 3' hydroxylase* (*F3'H*; Figure 1) which, in wild type tomato, is expressed in the peel during ripening, while only very low expression can be detected in the flesh [27]. The phenotype of the 'Crimson' tomatoes confirmed that the *F3'H* gene was expressed in the fruit peel but was not induced by *Del/Ros 1* in fruit flesh. The generation of tomatoes producing exclusively P1 would require the inactivation of *F3'H* in the peel, while tomatoes with high levels of C2 could be obtained by increasing the expression of *F3'H* in the flesh, since this gene is not a target of *Del/Ros1* [25]. The latter objective was achieved by fruit specific-expression of *AtMYB12*, which regulates the biosynthesis of flavonols [19] through transcriptional activation of biosynthetic genes, including *F3'H* [20]. This line (*AtMYB12*, here called 'Yellow'), the D3-producing 'Purple' line (*Del/Ros1*), and the progeny of their cross (*Del/Ros1; AtMYB12*, named 'Indigo') were available in both Micro Tom and Money Maker varieties. We used both genetic backgrounds in our breeding programme to generate P1 and C2 plants with big (Money Maker-like) or small (Micro Tom-like) fruit for further comparative studies; these lines still carry some of the genetic background of VF36 used to introduce the *f3'5'h* mutation. After crossing and selection of F2 plants, we obtained a new line (*Del/Ros1; AmDFR; AtMYB12; f3'5'h*) producing fruit with a novel deep dark shade of red that we named 'Magenta' and a higher content of anthocyanins compared to 'Pink' tomatoes (Figure 2).

A comparative analysis of the juice of 'Magenta', 'Indigo' and 'Crimson' in a Money Maker-like background is presented in Figure 3 and Table 1. 'Crimson' mainly produced P1, but low levels of C2 pigments could also be detected. The most abundant compound was identified as pelargonidin 3-(coumaroyl)-rutinoside-5-glucoside (also known as pelanin). 'Magenta' contained almost exclusively C2, with traces of pelanin. The main compound was peonidin 3-(coumaroyl)-rutinoside-5-glucoside (peonanin). The profile of 'Indigo' was identical to 'Purple', producing only D3 and showing petunidin 3-(coumaroyl)-rutinoside-5-glucoside (petanin) as the most prominent compound. Overall, the pattern of decoration (glycosylation, acylation, methylation) was remarkably similar in these three varieties and the differences were primarily associated with the level of hydroxylation of the B-ring, resulting in fruit and juices with distinctive colours.

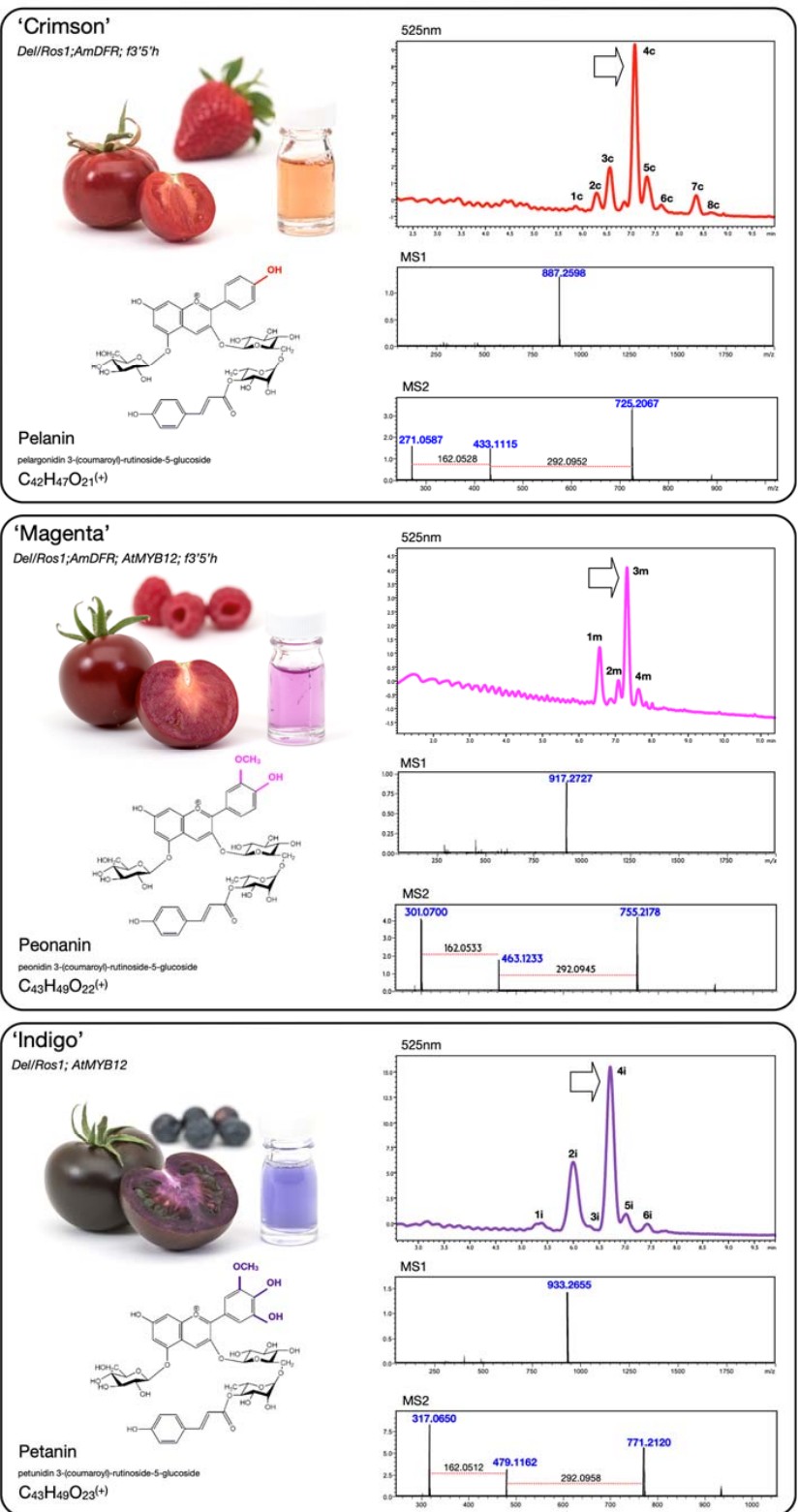

**Figure 3.** Fruit phenotypes and metabolic analyses of fruit juice extracted from 'Crimson', 'Magenta' and 'Indigo' tomatoes in a Money Maker-like background. For each variety, anthocyanins were analysed using HPLC, and MS1/MS2 spectra are provided for the most abundant peaks, indicated by arrows. Chemical structures, names and formulae of the corresponding compounds are reported. The glass vials contain tomato juice after centrifugation and dilution for better representation of juice colour. The anthocyanins (P1, C2 and D3) engineered in tomato mimic the three classes naturally produced in strawberries, raspberries and blueberries, illustrated in the background.

**Table 1.** Identification of the major anthocyanins in juice extracted from 'Crimson', 'Magenta' and 'Indigo' tomato fruit. Peak numbers correspond to those indicated in Figure 3. ESI-MS: electro spray ionization mass spectra; *m/z*: molecular mass of the compound.

| 'Crimson' | | | *Del/Ros1; AmDFR; f3′5′h* |
|---|---|---|---|
| **Peak** | **Rt (min)** | **ESI-MS (*m/z*)** | **Compound** |
| 1c | 5.86 | 919 | cyanidin 3-(caffeoyl)-rutinoside-5-glucoside |
| 2c | 6.29 | 903 | pelargonidin 3-(caffeoyl)-rutinoside-5-glucoside |
| 3c | 6.57 | 933 | peonidin 3-(caffeoyl)-rutinoside-5-glucoside |
| | | 903 | cyanidin 3-(coumaroyl)-rutinoside-5-glucoside |
| 4c | 7.08 | 887 | pelargonidin 3-(coumaroyl)-rutinoside-5-glucoside |
| 5c | 7.33 | 917 | pelargonidin 3-(feruloyl)-rutinoside-5-glucoside |
| | | 917 | peonidin 3-(coumaroyl)-rutinoside-5-glucoside |
| 6c | 7.63 | 947 | peonidin 3-(feruloyl)-rutinoside-5-glucoside |
| 7c | 8.35 | 725 | pelargonidin 3-(coumaroyl)-rutinoside |
| 8c | 8.7 | 755 | pelargonidin 3-(feruloyl)-rutinoside |
| 'Magenta' | | | *Del/Ros1; AmDFR; AtMYB12; f3′5′h* |
| **Peak** | **Rt (min)** | **ESI-MS (*m/z*)** | **Compound** |
| 1m | 6.57 | 933 | peonidin 3-(caffeoyl)-rutinoside-5-glucoside |
| | | 903 | cyanidin 3-(coumaroyl)-rutinoside-5-glucoside |
| 2m | 7.09 | 887 | pelargonidin 3-(coumaroyl)-rutinoside-5-glucoside |
| 3m | 7.32 | 917 | peonidin 3-(coumaroyl)-rutinoside-5-glucoside |
| 4m | 7.64 | 947 | peonidin 3-(feruloyl)-rutinoside-5-glucoside |
| 'Indigo' | | | *Del/Ros1; AtMYB12* |
| **Peak** | **Rt (min)** | **ESI-MS (*m/z*)** | **Compound** |
| 1i | 5.28 | 935 | delphinidin 3-(caffeoyl)-rutinoside-5-glucoside |
| 2i | 6 | 919 | delphinidin 3-(coumaroyl)-rutinoside-5-glucoside petunidin 3-(caffeoyl)-rutinoside-5-glucoside |
| | | 949 | |
| 3i | 6.32 | 949 | delphinidin 3-(feruloyl)-rutinoside-5-glucoside |
| 4i | 6.71 | 933 | petunidin 3-(coumaroyl)-rutinoside-5-glucoside |
| 5i | 7.02 | 963 | petunidin 3-(feruloyl)-rutinoside-5-glucoside |
| 6i | 7.43 | 947 | malvidin 3-(coumaroyl)-rutinoside-5-glucoside |

## 4. Discussion

Anthocyanin pigments play a fundamental role in the colour of flowers and fruit. The hydroxylation pattern of the B-ring is a major determinant, restricting colour to a wavelength spectrum that shifts from orange to blue as the number of hydroxyl groups increases. This simple chemical modification has a profound effect on the ecological roles of anthocyanins in terms of visual signals for pollination and seed dispersal. It is less clear, however, whether it can influence other functions of anthocyanins such as the ability to contribute to resistance against biotic and abiotic stresses or, as part of the normal diet, to provide protection against human chronic diseases. A major obstacle for these lines of research is the fact that plant species very rarely possess the ability to synthesise all three subclasses of pigments determined by the hydroxylation pattern of the B-ring.

From a horticultural perspective, selection of new colours and patterns is a major objective of ornamental plant breeding. As symbolised by the legendary 'blue rose'— mysterious, magical, and unattainable—many plant species are intrinsically unable to produce the purple/blue D3 pigments, such that the model plant Arabidopsis, and many important ornamentals, such as lily, rose, carnation, gerbera, and peony, all lack the *F3′5′H* gene that catalyses the hydroxylation at the 3′ and 5′ positions of the B-ring (Figure 1). Consequently, the elusive 'blue rose' as well as blue varieties of chrysanthemum and

carnation have been developed only through heterologous expression of *F3′5′H* genes isolated from petunia, pansy, or Canterbury bells [31]. Phylogenetic analyses have shown that *F3′5′H* evolved, probably on multiple occasions, from *F3′H* [28], which encodes a flavonoid hydroxylase ubiquitous in higher plants and responsible for the generation of C2, the most abundant subclass of anthocyanins (Figure 1). On the other hand, D3-producing plants with a functional F3′5′H are often unable to accumulate P1 pigments. This limitation has frequently been associated with the specificity of DFR, the first committed enzyme of the anthocyanin pathway, to reduce dihydroflavonols, particularly the ability to reduce dihydrokaempferol, the precursor of orange P1 pigments (Figure 1). For example, orange petunia flowers producing P1 could be developed only by introducing, in a suitable background, genes from maize or gerbera encoding DFR enzymes with broad substrate specificity [32,33]. In summary, in a particular plant species, the contribution of P1, C2 and D3 depends not only on the relative activity of F3′H and F3′5′H, but also on the substate preferences of DFR.

Regardless of the subclasses of anthocyanins produced, in higher plants, the biosynthetic pathway is activated by a conserved complex of transcription factors [34]. In tomato fruit, this regulation was lost in the ancestor of domesticated tomato, but can be reintroduced in fruit skin by introgression of regulatory genes from wild relatives [35] or, more effectively and in the whole fruit, by ectopic expression of two snapdragon regulatory genes [18]. In both cases, only D3-type anthocyanins were produced, reflecting the anthocyanin composition of vegetative tissues in wild type tomatoes and in other species of the Solanaceae family, such as aubergine and petunia, all containing functional *F3′5′H* genes.

To generate tomatoes producing the other two subclasses of anthocyanins, P1 and C2, we first introduced an *f3′5′h* mutation in *Del/Ros1* 'Purple' tomato. In this new 'Pink' variety (*Del/Ros1; f3′5′h*), D3 production was entirely lost and only C2, but not P1, could be detected. We concluded that the premature stop codon in F3′5′H caused a complete loss-of-function allele and that the anthocyanin biosynthetic machinery in wild type tomato is adapted to produce D3 preferentially, with reduced activity on C2 but no activity on P1 precursors. The visible presence of anthocyanins in 'Pink' tomatoes was surprising. The absence of anthocyanins in tomato fruit expressing two regulatory genes from maize (*LC* and *C1*) was attributed to their inability to activate *F3′5′H*, in combination with the strong preference of DFR for the tri-hydroxylated precursor dihydromyricetin [27]. Our data indicate that this interpretation is incomplete, since C2 can be produced in the absence of a functional *F3′5′H* gene. It is possible that, in *LC/C1* tomatoes, the expression of other genes may be insufficient, in particular the expression of those required for the transport into the vacuole, since all the other essential anthocyanin biosynthetic genes were reported to be upregulated. A distinctive feature of all the tomato lines expressing *Del/Ros1*, irrespective of the *f3′5′h* mutation, is the presence of a pigmented vascular tissue, which was not observed in *LC/C1* fruit. This phenotype may be due to the activity of a gene originally reported to encode a cinnamoyl CoA reductase (CCR), a key enzyme in lignin biosynthesis, which, interestingly shows remarkable homology with DFR [36]. In fact, when the *aw* mutation, which contains a non-functional *DFR* gene [37], was introduced in *Del/Ros1* tomatoes, we also observed anthocyanins in the fruit's vascular tissue, but not in any other fruit tissues (Figure S4). The analysis of 'Pink' (*Del/Ros1; f3′5′h*) fruit also suggest that, contrary to previous suggestions [27], the tomato DFR enzyme can recognise the di-hydroxylated precursor dihydroquercetin as a substrate. The low levels of anthocyanin in *Del/Ros1; f3′5′h* fruit compared to 'Purple' (*Del/Ros1*) fruit can be explained not only by reduced activity of the tomato DFR on dihydroquercetin, but also by the very low expression of *F3′H*, the hydroxylase required to produce this precursor, which is restricted to the fruit peel in wild type tomatoes. In 'Magenta' tomatoes, we were able to increase the levels of C2 8-fold by introducing a gene from snapdragon (*AmDFR*) encoding an enzyme with broader substrate specificity together with the transcription factor AtMYB12 which, unlike Del and Ros1, can induce *F3′H* expression [20].

The development of P1-producing tomatoes was more challenging, since this class of anthocyanins was completely undetectable in *Del/Ros1* fruit, regardless of the presence of a functional F3′5′H. These pigments could be produced only after introduction of the *AmDFR* gene from snapdragon, which is known to be able to synthesise both C2 and P1 type anthocyanins in flowers of *Antirrhinum majus* [38,39]. A short region in DFR has been associated with substrate recognition and specific amino acids residues that confer the ability to accept the P1 precursor dihydrokaempferol have been identified [30,40]; more recently, the presence of natural D3- or P1-producing accessions of *Lysimachia arvensis* has been attributed to the differential expression of two DFR genes encoding enzymes with differences in the substrate binding site [41]. An alignment of the 26 amino acids that form this region is provided in Figure S5, highlighting the role of a specific residue (asparagine in the enzymes able to accept dihydrokaempferol) considered to control substrate recognition. The expression of *AmDFR* in a *Del/Ros1; f3′5′h* background was able to shift the production of anthocyanins from C2 to P1 in tomato flesh but not in the peel, where only C2 could be detected. This is likely to be due to the natural expression of *F3′H* in this tissue [27], and is consistent with the preferential accumulation in the peel of rutin (quercetin glucoside), a flavonol which also requires *F3′H* for its biosynthesis. Our attempts to downregulate *F3′H* expression employing an RNAi construct driven by a 35S promoter resulted in sterile plants unable to set fruit even when used as female or male parents in crosses with other lines. A different strategy may involve the use of tomato mutants (*pf* and *y*), which are unable to accumulate flavonoids because of mutations affecting *SlMYB12*, the tomato orthologue of *AtMYB12* [42,43]. Consequently, the expression of all flavonoid biosynthetic genes, including *F3′H*, is considerably reduced. These mutants could be used to generate tomato fruit containing exclusively P1 due to the lack of expression of *F3′H* while the activation of the other anthocyanin biosynthetic genes would be ensured by the presence of Del/Ros1.

In terms of individual anthocyanins produced, the different combinations of genes present in 'Purple', 'Indigo', 'Magenta' and 'Crimson' tomato do not affect the general pattern of decoration of the anthocyanin structure, indicating that this is determined primarily by the activity of *Del/Ros1* (or the functionally equivalent transcription factors in tomato [44,45]) and endogenous tomato decorating enzymes. In all of the different lines, the most abundant modification involves glycosylation at position 5 of the A ring and the attachment of glucose and rhamnose followed by acylation with coumaric acid on the rutinoside at position 3 of the C ring (Figure 3) [25]. This substitution, 3-(coumaroyl)-rutinoside-5-glucoside, is the most abundant structure in Solanaceae [46]. Acylation with other organic acids (caffeic and ferulic) was also observed in equal proportion in all the tomato lines (Table 1). Obvious differences where present in the number of hydroxyl groups on the B-ring and in their level of methylation. The group at position 3′, absent in P1, is preferentially methylated in both C2 and D3 tomatoes. The methylation at position 5′, present only in D3 tomatoes, is poorly represented in the anthocyanin profile of 'Purple' and 'Indigo' fruit. This is the only notable difference between purple tomatoes obtained by metabolic engineering (*Del/Ros1*) or introgression (*Aft/Aft; atv/atv*), where methylation at both positions to generate malvidin anthocyanins is more frequent [26,47]. Since methylation at positions 3′ and 5′ is carried out by the same transferase OMT [48], this discrepancy could be explained by low expression of *OMT* in *Del/Ros1* tomatoes, thus increasing the ratio between unmethylated delphinidin and di-methylated malvidin. In both types of purple tomato, however, the mono-methylated petunidin derivative is the most abundant compound. The hydroxyl group at position 4′, present in all the 3′ classes of anthocyanins, is never methylated, consistent with the lack of in vitro activity of OMT at this position [48].

Overall, we have successfully engineered high levels of P1, C2 or D3 anthocyanins specifically in the flesh of tomato fruit. This is the first example of nearly isogenic plant material, available for comparative studies, where the three subclasses of anthocyanins are separately present. A limitation of our study is the absence of a tomato variety producing P1 in the peel. On the other hand, it is possible to easily produce large amounts of tomato

juice or extracts containing individual classes of anthocyanins without contamination from the peel. These extracts can be tested on human cell lines or used for animal feeding experiments to assess the effects of P1, C2 and D3 in different models and provide preliminary information for dietary intervention and disease prevention.

## 5. Conclusions

Fruit-specific expression of regulatory and biosynthetic genes, combined with the availability of a tomato mutant, resulted in the development of tomatoes producing high levels of three different classes of anthocyanins that confer distinctive fruit colours.

**Supplementary Materials:** The following are available online at https://www.mdpi.com/article/10.3390/horticulturae7090327/s1, Figure S1: Phenotypic and genetic analysis of tomato seedlings harbouring the *f3'f5'h* mutation. Figure S2: Analysis of anthocyanins extracted from the fruit of 'Pink' tomatoes (*Del/Ros1; f3'5'h*). Figure S3: Analysis of anthocyanins from the fruit of 'Crimson' tomatoes (*Del/Ros1; AmDFR; f3'5'h*). Figure S4: Fruit phenotypes of tomato lines expressing *Del/Ros1* in the presence of homozygous mutations affecting *F3'5'H* or *DFR*. Figure S5: Alignment of a region of 26 amino acids in DFR identified as the substrate binding site for dihydroflavonols. Table S1: Sequences of the primers used in the study.

**Author Contributions:** Conceptualization, C.M. and E.B.; investigation, K.B., L.H. and E.B.; writing—original draft preparation, E.B.; writing—review and editing, C.M.; visualization, E.B.; funding acquisition, C.M. All authors have read and agreed to the published version of the manuscript.

**Funding:** Research funded by the Biotechnological and Biological Scientific Research Council (BBSRC) through projects ERA-IB ANTHOPLUS project (031A336A0), the BBSRC-OpenPlant Synthetic Biology Research grant BB/L014130/1, the Institute Strategic Programs 'Understanding and Exploiting Plant and Microbial Secondary Metabolism' (BB/J004596/1) and 'Molecules from Nature' (BB/P012523/1). K.B. was supported by a BBSRC PhD scholarship.

**Institutional Review Board Statement:** Not applicable.

**Informed Consent Statement:** Not applicable.

**Data Availability Statement:** Not applicable.

**Acknowledgments:** The authors would like to thank Andrew Davies and Phil Robinson for providing photographs.

**Conflicts of Interest:** The authors declare no conflict of interest.

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
