# Peer review of "Beyond Purple Tomatoes: Combined Strategies Targeting Anthocyanins to Generate Crimson, Magenta, and Indigo Fruit"

_horticulturae, doi:10.3390/horticulturae7090327_

Round 1

Reviewer 1 Report

In the abstract, the authors said "based on fruit-specific engineering of three transcription factors and a single biosynthetic gene", what are the three TFs and a single biosynthetic gene? It should mention in the abstract and introduce in the introduction part. The result part includes lots of stuff should move to disscusion part.

Author Response

Response to Reviewer 1

In the abstract, the authors said "based on fruit-specific engineering of three transcription factors and a single biosynthetic gene", what are the three TFs and a single biosynthetic gene? It should mention in the abstract and introduce in the introduction part. The result part includes lots of stuff should move to disscusion part.

            The names of the genes have been included in the Abstract and Introduction.

We have moved and clarified a paragraph from Results to Discussion. We kept the initial introductory paragraph which is required to provide the rationale behind our approach, but the Result section is largely a description of the experiments involved.

Reviewer 2 Report

Very interesting approach to the research topic and remarkable results

Author Response

Response to Reviewer 2

English language and style

(x) Extensive editing of English language and style required
( ) Moderate English changes required
( ) English language and style are fine/minor spell check required
( ) I don't feel qualified to judge about the English language and style

Very interesting approach to the research topic and remarkable results

Many thanks for the kind comment. We are happy to correct and improve the English language, but we think that the wrong box may have been incorrectly selected in the form above.

Reviewer 3 Report

The article “Beyond Purple tomatoes: combined strategies targeting anthocyanins to generate Crimson, Magenta, and Indigo fruit” is a successful attempt to engineer tomatoes plants separately producing three different classes of plant pigment anthocyanin (cyanidin, pelargonidin and delphinidin) on nearly the same genetic background. These results are very valuable because such plants can be (and, I presume, surely will be) used in an experiment studying the health benefits of specific anthocyanin types. There is a number of evidences supporting positive role of anthocyanin on the health in the literature but we still lack complete understanding of how exactly specific anthocyanins can be involved in these processes, so, tomato lines producing major anthocyanin classes separately and in a quantity comparable to blueberry (‘Indigo’ line) are very valuable plant material for further research. The article is clearly written, illustrations are very picturesque and descriptive. Although I have a few questions and suggestions presented below, I think the manuscript can be accepted even in the present form.

Line 81: “although many berries accumulate high levels on anthocyaninsperhaps, “although many berries accumulate high levels of anthocyanins”.

Lines 147-148: “The desired combinations of transgenes were confirmed by DNA extraction and PCR analysis using gene specific primers. Please clarify which primers and DNA extraction method were used or include appropriate reference.

Lines 185-186: “Both approaches rely on the introduction on new regulatory genes” – perhaps, Both approaches rely on the introduction of new regulatory genes”.

Lines 197-199: “We crossed this mutant to the ‘Purple’ tomato line (Del/Ros1) expressing two regulatory genes from Antirrhinum majus (snap dragon) under the control of the fruit specific promoter E8 [18]. Please clarify which regulatory genes are present in the ‘Purple’ tomato line because it is not very convenient to search reference to understand it.

Lines 209-212: “We concluded that, while the premature stop codon in F3’5’H caused a complete loss of function allele, the tomato anthocyanin biosynthetic machinery is adapted to produce D3 anthocyanins preferentially, with reduced activity on C2 but no activity on P1 precursors.

According to the scheme on the Figure 1 the lack of F3’5’H activity should lead to “the tomato anthocyanin biosynthetic machinery is adapted to produce C2 anthocyanins preferentially, with reduced activity on P1 but no activity on D3 precursors” ?. Or did I miss something from the biosynthetic pathway?

Lines 215-216: “Figure 2. Summary of the phenotypes and genotypes of the varieties used in this study in Micro Tom or Micro Tom-like background”. There is no mention of Micro Tom or Micro Tom-like background in the Materials and Methods as well as Money Maker-like background (line 250).

Author Response

Response to Reviewer 3

The article “Beyond Purple tomatoes: combined strategies targeting anthocyanins to generate Crimson, Magenta, and Indigo fruit” is a successful attempt to engineer tomatoes plants separately producing three different classes of plant pigment anthocyanin (cyanidin, pelargonidin and delphinidin) on nearly the same genetic background. These results are very valuable because such plants can be (and, I presume, surely will be) used in an experiment studying the health benefits of specific anthocyanin types. There is a number of evidences supporting positive role of anthocyanin on the health in the literature but we still lack complete understanding of how exactly specific anthocyanins can be involved in these processes, so, tomato lines producing major anthocyanin classes separately and in a quantity comparable to blueberry (‘Indigo’ line) are very valuable plant material for further research. The article is clearly written, illustrations are very picturesque and descriptive. Although I have a few questions and suggestions presented below, I think the manuscript can be accepted even in the present form.

We appreciate the Comments and Suggestions of the Reviewer.

Line 81: “although many berries accumulate high levels on anthocyanins” — perhaps, “although many berries accumulate high levels of anthocyanins”.

We have corrected the typo.

Lines 147-148: “The desired combinations of transgenes were confirmed by DNA extraction and PCR analysis using gene specific primers”. Please clarify which primers and DNA extraction method were used or include appropriate reference.

We have included a comprehensive list of primers used in this study (Table S1) and appropriate references to it in the main text.

Lines 185-186: “Both approaches rely on the introduction on new regulatory genes” – perhaps, “Both approaches rely on the introduction of new regulatory genes”.

We have corrected the typo.

Lines 197-199: “We crossed this mutant to the ‘Purple’ tomato line (Del/Ros1) expressing two regulatory genes from Antirrhinum majus (snap dragon) under the control of the fruit specific promoter E8 [18]“. Please clarify which regulatory genes are present in the ‘Purple’ tomato line because it is not very convenient to search reference to understand it.

We have included the names of the genes in the sentence and specified that they encode bHLH- and MYB-type transcription factors.

Lines 209-212: “We concluded that, while the premature stop codon in F3’5’H caused a complete loss of function allele, the tomato anthocyanin biosynthetic machinery is adapted to produce D3 anthocyanins preferentially, with reduced activity on C2 but no activity on P1 precursors”.

According to the scheme on the Figure 1 the lack of F3’5’H activity should lead to “the tomato anthocyanin biosynthetic machinery is adapted to produce C2 anthocyanins preferentially, with reduced activity on P1 but no activity on D3 precursors” ?. Or did I miss something from the biosynthetic pathway?

We apologise for any confusion. Our comment refers to the anthocyanin biosynthetic machinery in wild type tomato, not in the ‘Pink’ tomato described in this section. We show that, in the presence of the f3’5’h mutation, tomato is still able to produce low levels of C2 (therefore, reduced activity on C2 precursors) but no detectable P1 (therefore, no activity on P1 precursors). We have specified that this conclusion refers to wild type tomato, changed the structure of the sentence and moved it to the Discussion section.

Lines 215-216: “Figure 2. Summary of the phenotypes and genotypes of the varieties used in this study in Micro Tom or Micro Tom-like background”. There is no mention of Micro Tom or Micro Tom-like background in the Materials and Methods as well as Money Maker-like background (line 250).

            We have clarified the use of different genetic backgrounds in Material and Methods (2.3. Tomato crosses and selection of plants) and specified the crossing, backcrossing and generations of self-pollination involved.

Round 2

Reviewer 1 Report

The manuscript can be accepted.